# Principal Graph Encoder Embedding and Principal Community Detection

## Abstract

In this paper, we introduce the concept of principal communities and propose a principal graph encoder embedding method that concurrently detects these communities and achieves vertex embedding. Given a graph adjacency matrix with vertex labels, the method computes a sample community score for each community, ranking them to measure community importance and estimate a set of principal communities. The method then produces a vertex embedding by retaining only the dimensions corresponding to these principal communities. Theoretically, we define the population version of the encoder embedding and the community score based on a random Bernoulli graph distribution. We prove that the population principal graph encoder embedding preserves the conditional density of the vertex labels and that the population community score successfully distinguishes the principal communities. We conduct a variety of simulations to demonstrate the finite-sample accuracy in detecting ground-truth principal communities, as well as the advantages in embedding visualization and subsequent vertex classification. The method is further applied to a set of real-world graphs, showcasing its numerical advantages, including robustness to label noise and computational scalability.

## 1 Introduction

Graph data has become increasingly popular over the past two decades. It plays a pivotal role in modeling relationships between entities across a wide array of domains, including social networks, communication networks, webpage hyperlinks, and biological systems (Girvan & Newman, 2002; Newman, 2003; Barabási & Oltvai, 2004; Boccaletti et al., 2006; Varshney et al., 2011; Ugander et al., 2011). Given $n$ vertices and $s$ edges, a binary graph can be represented by an adjacency matrix $\mathbf{A} \in \{0, 1\}^{n \times n}$, where $\mathbf{A}(i, j) = 1$ means there exists an edge between vertex $i$ and vertex $j$, and 0 otherwise. The high dimensionality of graph data, dictated by the number of vertices, often necessitates dimension reduction techniques for subsequent inferences.

Dimension reduction techniques applied to graph data are commonly referred to as graph embedding. Specifically, graph embedding transforms the adjacency matrix into a low-dimensional Euclidean representation per vertex. While many such techniques exist, two popular and theoretically sound methods are spectral embedding (Priebe et al., 2019) and node2vec (Grover & Leskovec, 2016), with asymptotic theoretical guarantees such as convergence to the latent position (Sussman et al., 2012) and consistency in community recovery (Zhang & Tang, 2024), under popular random graph models such as the stochastic block model and random dot graph model (Karrer & Newman, 2011; Zhao et al., 2012; Athreya et al., 2018). The resulting vertex embeddings facilitate a wide range of downstream inference tasks, such as community detection (Rohe et al., 2011; Gallagher et al., 2023), vertex classification (Tang et al., 2013; Mehta et al., 2021), and the analysis of multiple graphs and time-series data (Arroyo et al., 2021; Gallagher et al., 2021).

The scalability of spectral embedding is often a bottleneck due to its use of singular value decomposition, which can be time-consuming for moderate to large graphs. When vertex labels are available for at least part of the vertex set, a recent method called one-hot graph encoder embedding (Shen et al., 2023b), which can be viewed as a supervised version of spectral embedding, is significantly faster yet shares similar theoretical

properties, such as convergence to the latent positions. It also has several applications to weighted, multiple, and dynamic graphs (Shen et al., 2024b;a; Shen, 2024; Shen et al., 2024c), often exhibiting significantly better finite-sample performance over spectral embedding with a fraction of the time required.

Building upon the one-hot graph encoder embedding, this paper proposes a principal graph encoder embedding algorithm. The key addition is the introduction of a sample community score that ranks the importance of each community. The community score is then used to estimate a set of principal communities that contribute to the decision boundary for separating vertices of different communities. Due to the duality of community and dimensionality in the encoder embedding, the principal graph encoder embedding achieves further dimension reduction by restricting the embedding to the dimensions corresponding to the principal communities. The proposed algorithm maintains the same computational complexity as the original encoder embedding, making it significantly faster than other graph embedding techniques. Additionally, the reduced dimensionality enhances both the speed and robustness of subsequent inference, particularly in the presence of a large number of redundant or noisy communities.

To theoretically justify the sample algorithm, we provide a population characterization of the encoder embedding and principal communities. We prove, under a random Bernoulli graph model, that the principal graph encoder embedding preserves the conditional density of the label vector, making the proposed method Bayes optimal for vertex classification. Furthermore, under a regularity condition, we demonstrate that the proposed sample community score converges to a population community score, which equals zero if and only if the corresponding community is not a principal community.

Through comprehensive simulations and real-data experiments, we validate the numerical performance and theoretical findings through embedding visualization, ground-truth principal community detection, and vertex classification. The proposed method demonstrates excellent numerical accuracy, computational scalability, and robustness against noisy data. Theorem proofs are provided in the appendix.

## 2    The Main Method

In this section, we present the principal graph encoder embedding method for a given sample graph, followed by discussions on several practical issues such as normalization, sample community score threshold, and label vector availability.

### 2.1    Principal Graph Encoder Embedding

- **Input**: The graph adjacency matrix $\mathbf{A} \in \{0,1\}^{n \times n}$ and a label vector $\mathbf{Y} \in \{0,1,\ldots,K\}^n$, where 1 to $K$ represent known labels, and 0 is a dummy category for vertices with unknown labels.

- **Step 1**: Compute the number of known observations per class, i.e.,

$$n_k = \sum_{i=1}^{n} 1(\mathbf{Y}(i) = k)$$

for $k = 1, \ldots, K$.

- **Step 2**: Compute the matrix $\mathbf{W} \in [0,1]^{n \times K}$ as follow: for each vertex $i = 1, \ldots, n$, set

$$\mathbf{W}(i,k) = 1/n_k$$

if and only if $\mathbf{Y}(i) = k$, and 0 otherwise. Note that vertices with unknown labels are effectively assigned zero values, i.e., $\mathbf{W}(i,:)$ is a zero vector if $\mathbf{Y}(i) = 0$.

- **Step 3**: Compute the original graph encoder embedding through matrix multiplication:

$$\mathbf{Z} = \mathbf{AW} \in [0,1]^{n \times K}.$$

- **Step 4 (Normalization)**: Given $\mathbf{Z}$ from step 3, for each $i$ where $\|\mathbf{Z}(i, \cdot)\| > 0$, update the embedding as follows:

$$\mathbf{Z}(i, \cdot) = \frac{\mathbf{Z}(i, \cdot)}{\|\mathbf{Z}(i, \cdot)\|}.$$

- **Step 5 (Sample Community Score)**: Based on $\mathbf{Z}$ in step 4, for each $k \in [1, K]$, compute the sample community score as follows:

$$\hat{\lambda}(k) = \frac{\max_{l=1,\ldots,K}\{\hat{\mu}(k|l)\} - \min_{l=1,\ldots,K}\{\hat{\mu}(k|l)\}}{\max_{l=1,\ldots,K}\{\hat{\sigma}(\tilde{\mathbf{Z}}(k|l))\}},$$

where

$$\hat{\mu}(k|l) = \frac{\sum_{i=1,\ldots,n}^{\mathbf{Y}(i)=l} \mathbf{Z}(i,k)}{n_l}, \hat{\sigma}^2(k|l) = \frac{\sum_{i=1,\ldots,n}^{\mathbf{Y}(i)=l} \mathbf{Z}^2(i,k)}{n_l - 1} - \hat{\mu}^2(k|l),$$

then set the estimated principal communities as $\hat{D} = \{k \in [1, K] \text{ and } \hat{\lambda}(k) > \epsilon\}$ for a positive threshold $\epsilon$. The choice of $\epsilon$ is discussed in later subsection.

- **Step 6 (Principal Encoder)**: Denote the embedding limited to $\hat{D}$ in $\mathbf{Z}$ as $\mathbf{Z}^{\hat{D}}$. Then re-normalize each vertex embedding, i.e., for each $i$, set

$$\mathbf{Z}^{\hat{D}}(i, \cdot) = \frac{\mathbf{Z}(i, \hat{D})}{\|\mathbf{Z}(i, \hat{D})\|},$$

- **Output**: The original graph encoder embedding $\mathbf{Z}$, the principal graph encoder embedding $\mathbf{Z}^{\hat{D}}$, the sample community score $\{\hat{\lambda}(k)\}$, and the estimated set of principal communities $\hat{D}$.

Note that steps 1 to 3 compute the original graph encoder embedding method as described in Shen et al. (2023b), while the normalization in step 4 was employed in Shen et al. (2023a; 2024b). Therefore, the main contributions of this work lie in steps 5 and 6, where we compute the sample community score for each community, restrict the original embedding to the estimated principal communities, and re-normalize to yield the proposed principal graph encoder embedding.

It is important to note that $\mathbf{Z}^{\hat{D}}$ does not remove any observations from the embedding; rather, it only removes the $k$th dimension when community $k$ is not a principal community, i.e., $\mathbf{Z}^{\hat{D}} \in \mathbb{R}^{n \times |\hat{D}|}$. Every vertex, whether it is from a principal community or not, is always present in the final embedding $\mathbf{Z}^{\hat{D}}$.

## 2.2 On Normalization

Normalization, a well-known technique in many methods, ensures that all vertex embeddings have the same norm. In the context of graph encoder embedding, normalization projects the resulting sample embedding onto a unit sphere, which helps eliminate degree differences and often leads to improved separation among communities (Shen et al., 2023a). This is particularly beneficial for heterogeneous graphs, which are common in real-world data. In our case, normalization ensures that the sample community score behaves well empirically. This is because the sample community score involves the computation of sample expectations and variances, and normalized embeddings effectively exclude degree variance from these calculations.

## 2.3 The Community Score

The community score is designed to measure the importance of each community and serves as the basis for selecting the principal communities. Intuitively, in the original graph encoder embedding, the $k$th dimension can be interpreted as the average connectivity of the target vertex to all vertices in community $k$. Therefore, the proposed community score checks whether there is significant variability within dimension $k$, or equivalently, whether the connectivity from other communities to community $k$ is close to a constant or not. If

the connectivity is almost the same, the numerator will be relatively small, or in the extreme case, simply zero, indicating that community $k$ has no information in separating other communities.

A practical question is how large the score should be for a community to qualify as principal. One possible approach is to rank the community scores and decide a proper cut-off via cross-validation. In the presented algorithm, we opt for a faster approach using an adaptive threshold $\epsilon$ for cut-off. To determine this threshold, we employ the profile likelihood method from Zhu & Ghodsi (2006), a popular technique for selecting an elbow threshold given a vector. We choose the third elbow of all sample community scores, denoting it as $\epsilon_{\mathbf{A}}$, and then set $\epsilon = \max\{\epsilon_{\mathbf{A}}, 0.7\}$.

Empirically, the third elbow is very effective for large values of $n$ and $K$. However, for smaller to moderate values of $(n, K)$, the third elbow may be overly conservative. Through experimentation across various models and real datasets, it has been observed that principal communities typically have scores around or higher than 1, while redundant communities tend to have scores of no more than 0.5 for small $(n, K)$. As a result, we settled on the maximum of the third elbow and 0.7 as an empirical choice for $\epsilon$.

### 2.4 On Label Vector

Note that the given algorithm assumes knowledge of the label vector $\mathbf{Y} \in \{0, 1, \ldots, K\}^n$, which is at least partially known, where 0 denotes the dummy category of unknown labels. However, the method can also be used without the label vector. One could either use a random initialization and k-means to estimate the ground-truth labels (Shen et al., 2023a), or employ a direct label estimation algorithm such as Louvain, Leiden, or label propagation (Blondel et al., 2008; Traag et al., 2019; 2011; Raghavan et al., 2007) to estimate a label vector directly from the graph. In either scenario, one can compute the community score and principal encoder accordingly for any estimated label vector. The meaning of principal communities will pertain to the estimated labels, and the population theories in the next section still apply. Therefore, it suffices to assume a given label vector for the purpose of this paper, regardless of whether the label vector is ground-truth or estimated from some algorithms.

### 2.5 Computational Complexity

The computational complexity of the principal graph encoder embedding (P-GEE) is the same as the original graph encoder embedding (GEE), which is $O(nK + s)$, where $s$ represents the number of edges Shen et al. (2023b; 2024a). This is because neither the normalization nor the community score computation increases the overall complexity.

For instance, the method is capable of embedding a graph with $100,000$ vertices, 40 classes, and 10 million edges in under 10 seconds on a standard computer using MATLAB code. While the additional steps 5-6 in P-GEE may make it marginally slower than GEE, the reduced dimensionality of $\mathbf{Z}^{\hat{D}}$ can actually enhance its speed and scalability for subsequent tasks such as vertex classification. This is demonstrated in our real data experiments.

## 3 Population Definition and Supporting Theory

In this section, we characterize the population behavior of the method on random graph models. We begin by reviewing several popular random graph models, followed by the introduction of a random graph variable. We then define the population version of the principal community and the graph encoder embedding for this graph variable. This framework allows us to prove that the principal graph encoder embedding preserves the conditional density of the label vector. Additionally, we demonstrate that the sample community score converges to a population community score, which, under a regularity condition, equals zero if and only if the corresponding community is not a principal community. It is important to note that while the other sections focus on the sample method applied to sample graphs, everything in this section pertains to the population version of the method.

### 3.1 Existing Random Graph Models

**The Stochastic Block Model**

The standard stochastic block model (SBM) is a widely used graph model known for its simplicity and ability to capture community structures (Holland et al., 1983; Snijders & Nowicki, 1997; Karrer & Newman, 2011). Under SBM, each vertex $i$ is first assigned a class label $\mathbf{Y}(i) \in \{1, \ldots, K\}$. This label can either be predetermined or assumed to follow a categorical distribution with prior probabilities $\{\pi_k \in (0, 1], \sum_{k=1}^{K} \pi_k = 1\}$.

Given the vertex labels, the model independently generates each edge between vertex $i$ and another vertex $j \neq i$ using a Bernoulli random variable:

$$\mathbf{A}(i, j) \sim \text{Bernoulli}(B(\mathbf{Y}(i), \mathbf{Y}(j))).$$

Here, $B = [B(k, l)] \in [0, 1]^{K \times K}$ represents the block probability matrix, which serves as the parameters of the model. In a directed graph, the lower diagonal of the adjacency matrix is generated using the same distribution, while in an undirected graph, the lower diagonals are set to be equal to the upper diagonals. Note that the model does not have self-loops, meaning that $\mathbf{A}(i, i) = 0$. Additionally, whether the graph is directed or undirected does not affect the results discussed in this paper.

**The Degree-Corrected Stochastic Block Model**

The standard stochastic block model (SBM) generates dense graphs where all vertices within the same class have the same expected degrees. However, many real-world graphs are heterogeneous, with different vertices having varying degrees, and the graph can be very sparse. To accommodate this, the degree-corrected stochastic block model (DC-SBM) was introduced as an extension of SBM (Zhao et al., 2012).

In addition to the existing parameters of SBM, DC-SBM assigns a non-negative and bounded degree parameter $\theta_i$ to each vertex $i$. Given these degrees, the edge between vertex $i$ and another vertex $j \neq i$ is independently generated by:

$$\mathbf{A}(i, j) \sim \text{Bernoulli}(\theta_i \theta_j B(\mathbf{Y}(i), \mathbf{Y}(j))).$$

When all degrees are set to 1, DC-SBM reduces to the standard SBM. Typically, degrees may be assumed to be fixed a priori or independently and identically distributed within each community. These degree parameters allow DC-SBM to better approximate real-world graphs.

**The Random Dot Product Graph**

Under the random dot product graph (RDPG), each vertex $i$ is associated with a hidden latent variable $U_i \overset{i.i.d.}{\sim} f_U \in \mathbb{R}^m$ (Young & Scheinerman, 2007; Athreya et al., 2018). Then each edge is independently generated as follows:

$$\mathbf{A}(i, j) \sim \text{Bernoulli}(< U_i, U_j >),$$

where $< \cdot, \cdot >$ denotes the inner product. To enable communities under RDPG, it suffices to assume the latent variable follows a K-component mixture distribution. In other words, each vertex is associated with a class label $\mathbf{Y}(i)$ such that

$$U_i | (\mathbf{Y}(i) = k) \overset{i.i.d.}{\sim} f_{U|k}.$$

### 3.2 Defining a Graph Variable

To characterize the graph embedding using a framework similar to the conventional setup of predictor and response variables, we formulate the above graph models into the following graph variable, called the random Bernoulli graph distribution.

**Definition 1.** *Given a vertex, we assume $Y$ is the underlying label that follows a categorical distribution with prior probabilities $\{\pi_k \in (0,1], \sum_{k=1}^{K} \pi_k = 1\}$. Additionally, $X \in \mathbb{R}^p$ is the latent variable with a $K$-component mixture distribution, denoted as*

$$X \sim \sum_{k=1}^{K} \pi_k f_{X|Y=k},$$

*where $f_{X|Y=k}$ represents the conditional density.*

*Moreover, we assume a known label vector $\vec{\mathcal{V}} = \{v_1, v_2, \ldots, v_m\} \in [1, K]^m$, where each $k \in [1, K]$ is present in $\vec{\mathcal{V}}$. Additionally, there exists a corresponding random matrix*

$$\vec{\mathcal{U}} = [U_1; U_2; \cdots; U_m] \in \mathbb{R}^{m \times p},$$

*where each $U_j$ is independently distributed with density $f_{X|Y=v_j}$.*

*We then define an $m$-dimensional random variable $A$ following the random Bernoulli graph distribution as*

$$A \sim RBG(X, \vec{\mathcal{V}}, \delta) \in \{0, 1\}^m,$$

*if and only if each dimension $A_j$ is distributed as*

$$A_j \sim Bernoulli(\delta(X, U_j)), j = 1, \ldots, m.$$

*Here, $\delta(\cdot, \cdot) : \mathbb{R}^p \times \mathbb{R}^p \to [0, 1]$ can be any deterministic function, such as weighted inner product or kernel function.*

Note that $\vec{\mathcal{V}}$ is a known vector. Alternatively, one could view it as independent sample realizations using the same categorical distribution of $Y$. As we have required each integer from 1 to $K$ to be present in $\vec{\mathcal{V}}$, it necessarily implies $m \geq K$.

In essence, the random Bernoulli graph distribution is a multivariate concatenation of mixture Bernoulli distributions. In this distribution, the probability of each Bernoulli trial is determined by a function involving the latent variable $X$ and an independent copy $U_j$ with a known label $v_j$. The random Bernoulli graph distribution is a versatile framework encompassing SBM, DC-SBM, RDPG, and more general cases, due to its flexibility in allowing any $\delta(\cdot, \cdot)$ and any particular distribution for $X$.

Consider the sample adjacency matrix and the labels $(\mathbf{A}, \mathbf{Y}) \in \{0, 1\}^{n \times n} \times \{1, 2, \ldots, K\}^n$ as an example, where the graph has no self-loop. Then, for each $i = 1, \ldots, n$, the $i$th row of $\mathbf{A}$ is distributed as

$$\mathbf{A}(i, :) \sim RBG(X, \vec{\mathcal{V}}, \delta),$$

where $X$ is the underlying latent variable for vertex $i$, and $\vec{\mathcal{V}}$ is the known sample labels of all other vertices. Note that the dimension $m = n - 1$ because $\mathbf{A}(i, i) = 0$, and it suffices to consider the edges between vertex $i$ and all other vertices.

### 3.3 The Principal Graph Encoder Embedding for the Graph Variable

In this section, we characterize the population version of the original graph encoder embedding, the principal community, and the principal graph encoder embedding on the graph variable. Note that their sample notations are $\mathbf{Z}$, $\hat{D}$, and $\mathbf{Z}^{\hat{D}}$ respectively in Section 2, and the corresponding population notations are $Z$, $D$, and $Z^D$ respectively in this section.

**Definition 2.** *Given a random graph variable $A \sim RBG(X, \vec{\mathcal{V}}, \delta)$. For each $k = 1, \ldots, K$, calculate*

$$m_k = \sum_{j=1}^{m} 1(v_j = k),$$

*where $1(v_j = k)$ equals 1 if $v_j = k$, and 0 otherwise.*

*We then compute the matrix $W \in \mathbb{R}^{m \times K}$ as follows:*

$$W(i,j) = \begin{cases} 1/m_k & \text{when } v_j = k, \\ 0 & \text{otherwise.} \end{cases}$$

*The population graph encoder embedding is then defined as $Z = AW \in [0,1]^K$.*

Note that the $W$ matrix is conceptually similar to the one-hot encoding scheme, except the entries are normalized rather than binary. Next, we introduce the concepts of principal and redundant communities for the graph variable:

**Definition 3.** *Given $A \sim RBG(X, \vec{\mathcal{V}}, \delta)$, and $U^k$ as an independent variable distributed as $f_{X|Y=k}$. A community $k$ is defined as a principal community if and only if*

$$Var\left(E(\delta(X, U^k) \mid X)\right) > 0.$$

*On the other hand, any community for which the above variance equals 0 is referred to as a redundant community.*

For example, in the stochastic block model, the condition $Var(E(\delta(X, U^k) \mid X)) = 0$ is equivalent to the $k$th column of the block probability matrix $B(:, k)$ being a constant vector, which does not provide any information about $Y$ via the edge probability. Finally, the principal graph encoder embedding can be defined as follows:

**Definition 4.** *Define $D$ as the set of principal communities, and $Z^D$ as the graph encoder embedding whose dimensions are restricted to the indices in $D$. We call $Z^D$ the principal graph encoder embedding.*

For example, if $K = 5$ and $D = \{1, 2\}$, then $Z$ spans five dimensions while $Z^D$ only keeps the first two dimensions from $Z$. The principal graph encoder embedding achieves additional dimension reduction compared to the original graph encoder embedding. Given the population definitions, the sample versions $\mathbf{Z}$, $\hat{D}$, and $\mathbf{Z}^{\hat{D}}$ in Section 2 can be viewed as sample estimates for the population counterparts $Z$, $D$, and $Z^D$.

### 3.4 Conditional Density Preserving Property

Based on the population setting, we can prove the principal graph encoder embedding preserves the conditional density, and as a result, preserves the Bayes optimal classification error via the classical pattern recognition framework (Devroye et al., 1996).

**Theorem 1.** *Given $A \sim RBG(X, \vec{\mathcal{V}}, \delta)$, the principal graph encoder embedding preserves the following conditional density:*

$$Y|A \stackrel{dist}{=} Y|Z \stackrel{dist}{=} Y|Z^D.$$

*Denote $L^*(Y, A)$ as the Bayes optimal error to classify $Y$ using $A$, we have*

$$L^*(Y, A) = L^*(Y, Z) = L^*(Y, Z^D).$$

Intuitively, the graph variable $A$ is an $m$-dimensional multivariate concatenation of mixture Bernoulli distributions, the original graph encoder embedding $Z$ is a $K$-dimensional multivariate concatenation of mixture Binomial distributions, and the principal graph encoder embedding $Z^D$ discards every dimension in $Z$ whose Binomial mixture component is equivalent to a single Binomial.

Note that this property is on the population level. For the sample version, we expect the property to hold for sufficiently large vertex size, rather than at any $n$, due to sample estimation variance. Moreover, the property does not imply that any classifier can be asymptotically optimal for the sample embedding. Only

when using the theoretical optimal Bayes classifier on the embedding $Z^D$ will the resulting optimal error be the same as the theoretical optimal error using the original graph variable $A$.

This theorem shows that the principal communities are well-defined, and retaining the dimensions corresponding to the principal communities is sufficient for subsequent vertex classification. While both the original graph encoder embedding and the principal graph encoder embedding are equivalent in population, the principal graph encoder embedding has fewer dimensions and therefore usually provides a finite-sample advantage in subsequent inference.

### 3.5 Population Community Score

While Theorem 1 establishes that the principal community is well-defined and preserves the conditional density, it remains to demonstrate that the proposed community score can effectively detect such principal communities. To this end, we first introduce the population community score:

**Definition 5.** *Define*

$$\lambda(k) = \frac{\max_{l=1,\ldots,K}\{E(Z_k|Y=l)\} - \min_{l=1,\ldots,K}\{E(Z_k|Y=l)\}}{\max_{l=1,\ldots,K}\{\sqrt{Var(Z_k|Y=l)}\}} \in [0, +\infty)$$

*as the population community score for each community $k \in [1, K]$.*

Since the sample community score used in Step 5 of Section 2.1 relies on the sample expectation and variance, which converge to their respective population counterparts, it follows immediately that

$$\hat{\lambda}(k) \overset{n\to\infty}{\to} \lambda(k).$$

In other words, the sample community score converges to the population community score.

The following theorem proves that, under a regularity condition, the population community score perfectly separates principal communities from redundant communities.

**Theorem 2.** *Assume $A \sim RBG(X, \vec{\mathcal{V}})$, and $\delta(X, U^k)|Y$ is independent of $X|Y$, which is satisfied under the stochastic block model. Then the population community score $\lambda(k) = 0$ if and only if community $k$ is a redundant community.*

Note that the condition $\delta(X, U^k)|Y$ is independent of $X|Y$ can also hold for the degree-corrected stochastic block model, as shown in the proof. Since DC-SBM is often a realistic model for many real-world graphs, we can expect the designed community score to perform well in practice. It is important to emphasize that this property holds at the population level, meaning it is expected to perform well for sufficiently large vertex sizes rather than any finite $n$, due to sample estimation variance.

Finally, there exist other alternative statistics that can consistently detect principal communities. For example, as shown in the proof, one could use the numerator of the community score or the variance of $E(Z_k|Y=l)$, both of which also equal zero under the same condition. Nevertheless, the numerator based on order statistics makes it more robust, and the denominator provides effective normalization for ranking and thresholding purposes, making the proposed community score well-behaved and robust in empirical assessments.

## 4 Simulations

We consider three simulated models with $K = 20$ and increasing $n$. In each model, vertex label assignment is randomly determined based on prior probabilities: $\pi_k = 0.25$ for $k = \{1, 2, 3\}$, then equally likely to be $0.25/(K-3)$ for the remaining classes. Given these labels, edge probabilities are generated under each model. Here are the details of the model parameters for each:

- **SBM**: Block probability matrix: $B(k, k) = 0.2$ for $k = \{1, 2, 3\}$, and $B(k, l) = 0.1$ otherwise.

- **DC-SBM**: Vertex degree generation:

$$\theta_i | (\mathbf{Y}(i) = y) \sim \text{Beta}(1, 5 + y/5).$$

Block probability matrix: $B(1,1) = 0.9$, $B(2,2) = 0.7$, $B(3,3) = 0.5$, and $B(k,l) = 0.1$ otherwise.

- **RDPG**: Latent variable $U \in \mathbb{R}^4$. For $k = \{1, 2, 3\}$,

$$U(:,k) | (Y = k) \sim \text{Uniform}(0.2, 0.3).$$

For $k > 3$,

$$U(:,k) | (Y > 3) \sim \text{Uniform}(0.1, 0.2).$$

For all dimensions $l \neq k$.

$$U(:,l) | (Y = k) \sim \text{Uniform}(0, 0.1).$$

In all three models, the parameter settings are designed such that vertices from the top three communities can be perfectly separated on a population level, and these communities are considered the principal communities, represented as $D = \{1, 2, 3\}$. On the other hand, vertices from the remaining communities are intentionally designed to be indistinguishable from each other, constituting redundant communities. As a result, 75% of the vertices belong to the principal communities and can be perfectly separated with a large sample size, while the remaining 25% of the vertices cannot be distinguished from each other.

## 4.1 Embedding Visualization and Sample Community Score

Figure 1 shows the adjacency matrix heatmap for one sample realization, the visualization of the resulting principal graph encoder embedding, and the sample community score for each dimension. The first column shows the heatmap of the generated adjacency matrix $\mathbf{A}$. The sample indices are sorted by class to highlight the block structure. All three graphs exhibit a similar block structure, with higher within-class probabilities for the top three communities. The SBM graph is the most dense graph, followed by RDPG, and the DC-SBM graph is the most sparse by design.

The second column presents the sample community scores $\hat{\lambda}(k)$ based on the proposed sample method. Clearly, the sample scores for the first three communities / dimensions stand out and are significantly higher than the others. As a result, the proposed method successfully identifies and reveals the ground-truth dimension, setting $\hat{D} = D = \{1, 2, 3\}$.

The third column visualizes the principal graph encoder embedding $\mathbf{Z}^{\hat{D}}$. Each dot represents the embedding for a vertex, and different colors indicate the class membership of each vertex, particularly those from the principal communities. Since $\hat{D} = D = \{1, 2, 3\}$, the embedding is in 3D and occupies the top three dimensions. We observe that the encoder embedding effectively separates the top three communities, while all redundant communities are mixed together and cannot be distinguished, which aligns with the given models.

## 4.2 Detection Accuracy and Vertex Classification

Using the same simulation models, we further assess the capability of the proposed method to identify the ground-truth dimensions and evaluate the quality of the embedding through a classification task on the vertex embedding. For each model, we generate sample graphs with increasing $n$, compute the sample community score, report accuracy in detecting the ground-truth principal communities, compute the principal graph encoder embedding (using training labels only via 5-fold evaluation), apply linear discriminant analysis as the classifier, and report the testing error on the testing vertices. This process is repeated for 100 Monte-Carlo replicates for each $n$, ensuring that all standard deviations fall within a margin of 1%. The average results are reported in Figure 2.

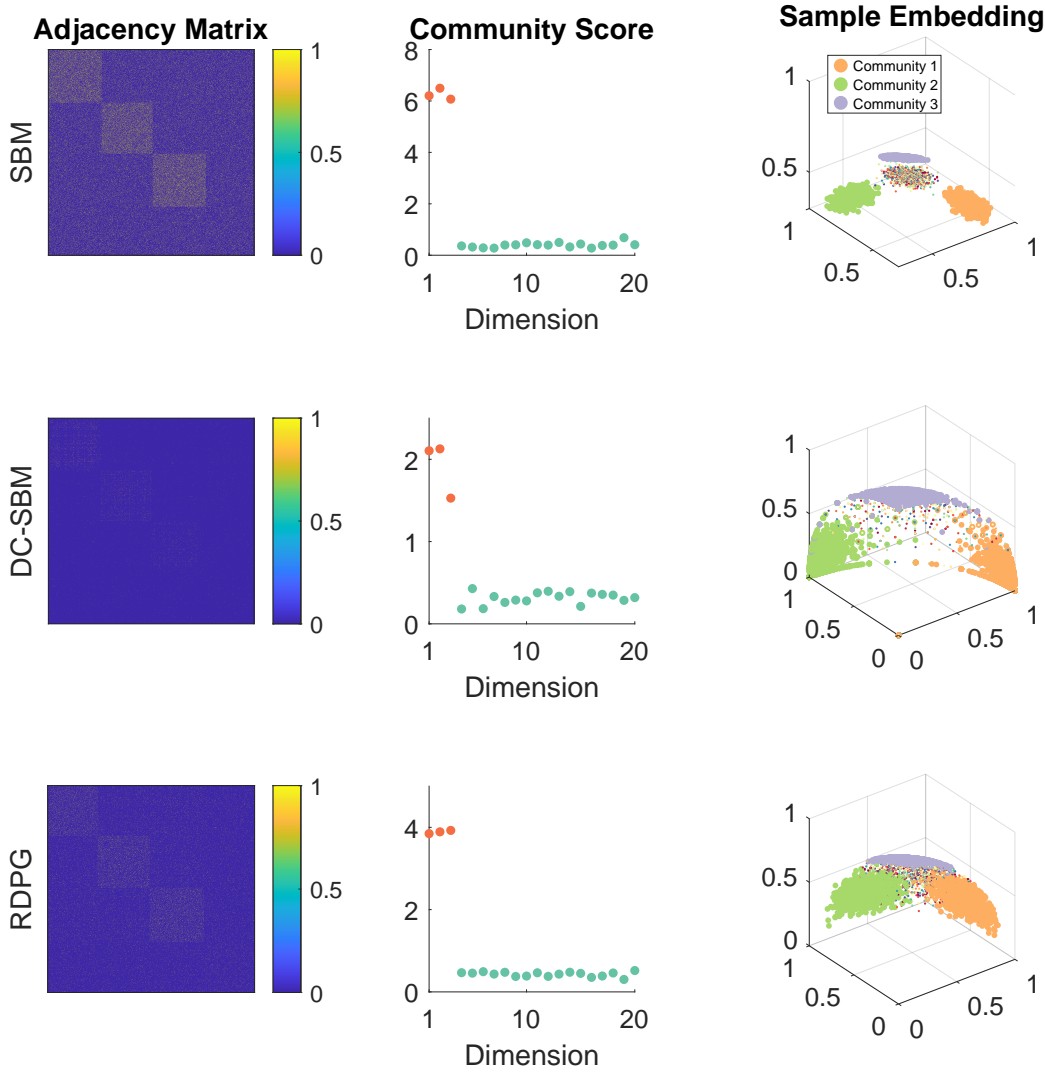

Figure 1: This figure visualizes the sample adjacency matrix for each model at $n = 5000$ and $K = 20$, the sample community scores for $k \in [1, 20]$, and the principal graph encoder embedding.

The first column of Figure 2 shows the sample community scores as $n$ increases. The red line represents the average sample community score among the principal communities, while the blue line represents the average sample community score among the redundant communities. For all three models, as $n$ increases, the principal communities and the redundant communities become increasingly separated in the sample scores.

This separation translates well to the second column of Figure 2, which shows that the sample algorithm quickly achieves a true positive rate of 1 and a false positive rate of 0 in detecting the true principal communities. In other words, $\text{Prob}(\hat{D} = D) \to 1$ for sufficiently large $n$. This implies that our method is consistent in detecting the ground-truth principal communities.

The third column of Figure 2 evaluates the quality of the graph embedding by conducting a 5-fold vertex classification task on the sample embedding. Specifically, we divide the vertices into 5 folds and test each fold individually. In each instance, we compute the sample embedding by setting all testing labels to 0, apply linear discriminant analysis to the embedding and labels of the training vertices, predict the testing labels using the embedding of the testing vertices, and then calculate the error by comparing the predicted label to the true testing label. According to the population model, we can compute that the optimal Bayes error is approximately 0.235 across all three models.

As the sample size increases, we observe that the classification error using the principal graph encoder embedding converges to the Bayes optimal error, as does the encoder embedding without principal community detection. This phenomenon supports the theorem that the encoder embedding indeed preserve the conditional density. While both the original embedding and the principal version converge to the optimal error, the principal version appears to perform slightly better across all settings. This improvement is due to reduced dimension while preserving the label information, which benefits the sample classification.

## 5 Real Data

### 5.1 Setting

We collected a diverse set of real graphs with associated labels from various sources, including the Network Repository[1] (Rossi & Ahmed, 2015), Stanford Network Data[2], and internally collected graph data.

Since the ground-truth principal communities are unknown in real graphs, we primarily use vertex classification on embedding as a proxy to evaluate the embedding quality. We compare this to the original graph encoder embedding to assess the quality of the principal community detection. We use 5-fold cross-validation and linear discriminant analysis, and compare the graph encoder embedding (GEE), the principal graph encoder embedding (P-GEE), the adjacency spectral embedding (ASE), and node2vec (N2V). ASE requires an explicit dimension choice, which is set to $d = 30$. For node2vec, we use the graspy package (Chung et al., 2019) with default parameters and 128 dimensions. Any directed graph was transformed to undirected, and any singleton vertex was removed.

### 5.2 Using Original Data

Table 1 summarizes the average error and standard deviation after conducting 100 Monte Carlo replicates for each given graph. It also provides basic dataset details, including $n$, $K$, and the median dimension choice $|\hat{D}|$ for P-GEE. The numerical results clearly indicate that both GEE and P-GEE deliver excellent performance across all datasets, outperforming spectral embedding in all cases and node2vec in most cases. We also observe that the proposed principal graph encoder embedding is generally very close to the original graph encoder embedding in classification error: by detecting and only retaining the dimensions corresponding to the principal communities, the principal GEE either maintains or slightly improves the classification error compared to the original GEE throughout all real data (except the IIP data with $K$ being only 3).

This implies that as long as $K$ is not too small, the principal GEE successfully extracts important communities that preserve sufficient label information and improves the classification error, consistent with the numerical behavior observed in the simulations. Another observation is that the encoder embedding produces the best error in most cases, and in the two cases where node2vec yields better error than GEE, GEE is also very close in error, suggesting its overall satisfactory performance.

### 5.3 Using Noisy Data

To further demonstrate the advantage and robustness of P-GEE, we conducted a noisy data experiment. We used the same real data and evaluation as above, with the exception that the vertex labels were partially polluted. Specifically, for each replicate, we randomly assigned 10% of the ground-truth vertex labels to one

---

[1]http://networkrepository.com/
[2]https://snap.stanford.edu/

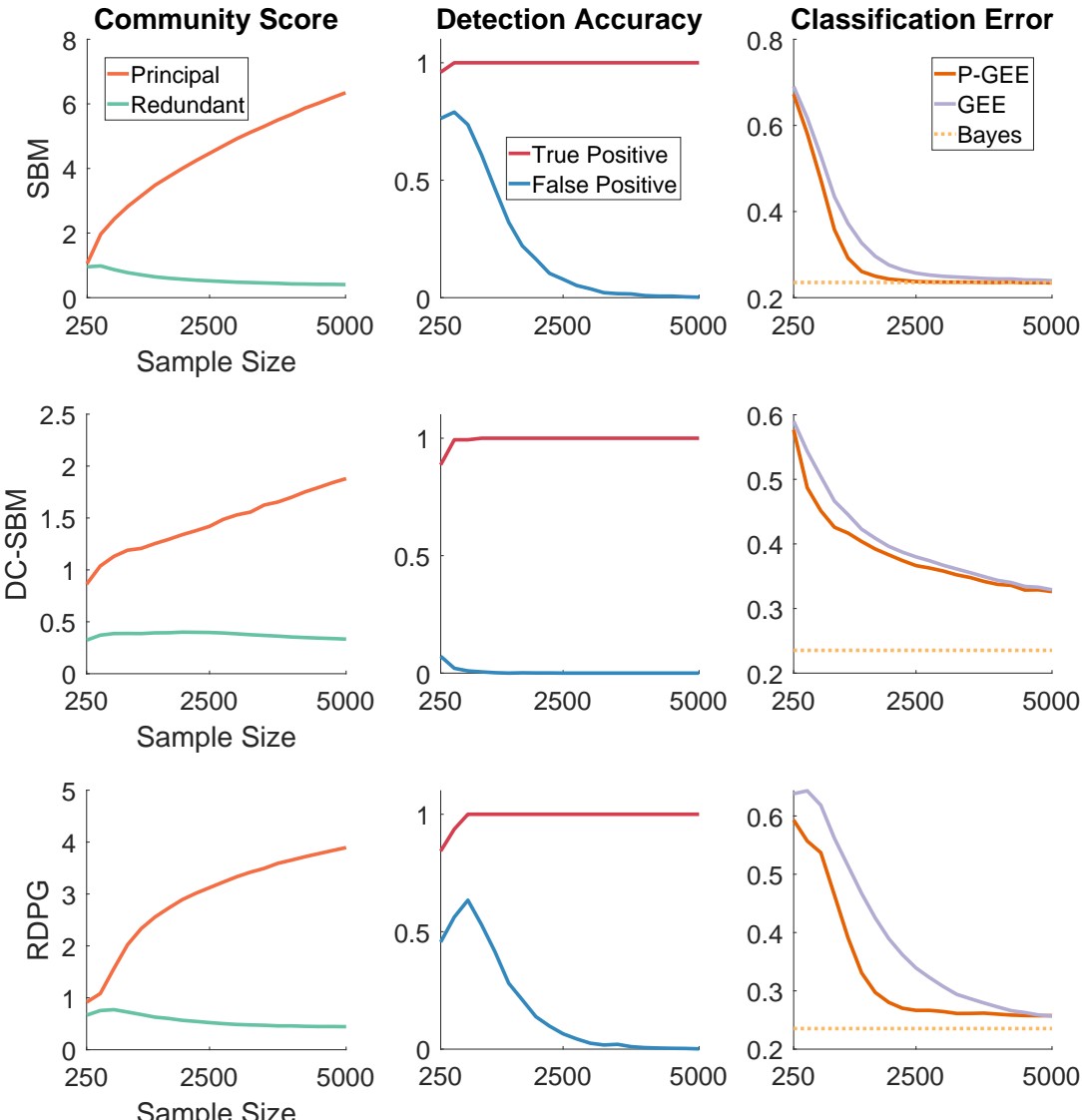

Figure 2: This figure displays the average sample community score, the average principal community detection accuracy, and the average vertex classification error using the embedding, based on 100 Monte-Carlo replicates with increasing $n$. P-GEE denotes the principal graph encoder embedding, and GEE denotes the original graph encoder embedding.

of 30 additional noise classes. For example, suppose $K = 3$ and vertex 1 belongs to class 2, so $\mathbf{Y}(1) = 2$. If vertex 1 is not polluted, we have $\mathbf{Y}^{noise}(1) = \mathbf{Y}(1) = 2$; otherwise, $\mathbf{Y}^{noise}(1) \in [4, 5, \ldots, 33]$ with equal probability.

We then used the given graph and noisy labels $\mathbf{Y}^{noise}$ to perform vertex embedding and classification for GEE, P-GEE, and ASE, and reported the results in Table 2. Comparing vertex classification accuracy, we observed that P-GEE consistently outperforms GEE and ASE in most cases. In fact, P-GEE is nearly insusceptible to label noise, achieving ideal error rates on the noisy data in most cases. Here, "ideal error"

|  | $(n, K)$ | GEE (%) | P-GEE (%) | $|\hat{D}|$ | ASE (%) | N2V (%) |
|---|---|---|---|---|---|---|
| Citeseer | $(3312, 6)$ | $32.8 \pm 0.6$ | $\mathbf{32.3} \pm 0.6$ | 4 | $60.3 \pm 0.5$ | $77.5 \pm 0.5$ |
| Cora | $(2708, 7)$ | $\mathbf{20.9} \pm 1.5$ | $\mathbf{20.9} \pm 1.5$ | 5 | $31.8 \pm 0.6$ | $75.1 \pm 0.5$ |
| Email | $(1005, 42)$ | $34.1 \pm 0.8$ | $34.2 \pm 0.8$ | 39 | $43.6 \pm 0.4$ | $\mathbf{29.2} \pm 0.5$ |
| IIP | $(219, 3)$ | $\mathbf{31.7} \pm 1.9$ | $32.7 \pm 1.7$ | 2 | $35.6 \pm 0.4$ | $48.9 \pm 3.2$ |
| IMDB | $(19503, 3)$ | $\mathbf{1.4} \pm 2.9$ | $\mathbf{1.4} \pm 2.9$ | 3 | $60.1 \pm 0.4$ | $44.8 \pm 0.1$ |
| LastFM | $(7624, 18)$ | $20.3 \pm 0.3$ | $20.3 \pm 0.3$ | 17 | $43.3 \pm 0.4$ | $\mathbf{14.7} \pm 0.1$ |
| Letter | $(10507, 15)$ | $\mathbf{7.4} \pm 0.3$ | $\mathbf{7.4} \pm 0.3$ | 4 | $89.2 \pm 0.3$ | $74.9 \pm 0.3$ |
| Phone | $(1703, 71)$ | $30.1 \pm 0.8$ | $\mathbf{28.6} \pm 0.8$ | 53 | $55.9 \pm 0.2$ | $83.7 \pm 0.5$ |
| Protein | $(43471, 3)$ | $\mathbf{30.8} \pm 0.2$ | $\mathbf{30.8} \pm 0.2$ | 3 | $51.0 \pm 0.7$ | $45.8 \pm 0.1$ |
| Pubmed | $(19717, 3)$ | $\mathbf{22.6} \pm 0.2$ | $\mathbf{22.6} \pm 0.2$ | 3 | $35.5 \pm 0.7$ | $58.9 \pm 0.2$ |

Table 1: Vertex classification error using 5-fold linear discriminant analysis on each graph embedding. The table reports the average error and standard deviation after 100 Monte Carlo replicates, highlighting the best error within each dataset. All accuracy are in percentile.

can be defined as the best error on the original data in the corresponding row of Table 1, plus approximately 10%. For example, on the IMDB data, the best error on the original data is 1%, while P-GEE achieves an error of 10.2% on the noisy data, compared to a much worse error of 28.3% for GEE on the noisy data. Similarly, on the PubMed data, the best error on the original data is 22.6%, with P-GEE achieving an error of 32.3% on the noisy data, while GEE on the noisy data has a higher error of 39.0%.

We also observed that the estimated number of principal communities $|\hat{D}|$ accurately matches the true $K$ of the original data in most cases, indicating that the sample community score remains robust and well-behaved in the presence of noisy data.

Finally, we compared the running times of GEE, P-GEE, and ASE, including both embedding and classification time. While the running times for P-GEE and GEE are mostly similar, the reduced dimensionality in P-GEE speeds up subsequent classification, leading to noticeable improvements in running time. Both GEE and P-GEE are significantly faster than spectral embedding for moderate to large graph sizes. For example, for relatively larger graphs, such as LastFM, letter, and protein data, which have tens of thousands of vertices, spectral embedding requires several seconds, whereas GEE only takes a fraction of a second.

|  | GEE (%) | Time (s) | P-GEE (%) | Time (s) | $(K, |\hat{D}|)$ | ASE (%) | Time (s) |
|---|---|---|---|---|---|---|---|
| Citeseer | $49.9 \pm 0.6$ | 0.11 | $\mathbf{41.5} \pm 0.5$ | 0.09 | $(6, 6)$ | $72.9 \pm 0.3$ | 0.16 |
| Cora | $42.6 \pm 0.6$ | 0.12 | $\mathbf{29.4} \pm 0.5$ | 0.11 | $(7, 7)$ | $61.8 \pm 0.4$ | 0.15 |
| Email | $\mathbf{50.5} \pm 0.8$ | 0.38 | $51.2 \pm 0.9$ | 0.37 | $(42, 60)$ | $50.8 \pm 0.5$ | 0.42 |
| IIP | $42.8 \pm 2.0$ | 0.02 | $\mathbf{39.2} \pm 2.0$ | 0.02 | $(3, 2)$ | $52.3 \pm 2.2$ | 0.03 |
| IMDB | $28.3 \pm 0.4$ | 0.17 | $\mathbf{10.2} \pm 0.05$ | 0.13 | $(3, 3)$ | $69.0 \pm 0.06$ | 0.75 |
| LastFM | $42.1 \pm 0.4$ | 0.22 | $\mathbf{30.4} \pm 0.3$ | 0.19 | $(18, 17)$ | $53.0 \pm 0.2$ | 5.9 |
| Letter | $29.7 \pm 0.3$ | 0.20 | $\mathbf{21.1} \pm 0.3$ | 0.18 | $(15, 15)$ | $90.8 \pm 0.03$ | 1.1 |
| Phone | $48.8 \pm 1.0$ | 0.75 | $\mathbf{41.3} \pm 0.9$ | 0.67 | $(71, 53)$ | $64.1 \pm 0.3$ | 0.70 |
| Protein | $42.5 \pm 0.2$ | 0.30 | $\mathbf{38.3} \pm 0.2$ | 0.17 | $(3, 3)$ | $56.6 \pm 0.02$ | 2.4 |
| Pubmed | $39.0 \pm 0.3$ | 0.16 | $\mathbf{32.3} \pm 0.2$ | 0.12 | $(3, 3)$ | $44.9 \pm 0.5$ | 0.48 |

Table 2: The evaluation is the same as in Table 1, except that 10% of the given labels are randomized into one of 30 noise groups. We also report the average running time. The best error within the noise columns is highlighted. All accuracy are presented as percentages, and all running times are in seconds.

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

# APPENDIX

To facilitate the proof, we introduce the following notations for conditioning and density arguments:

- We use $\cdot|\vec{\mathcal{U}}$ to denote the conditioning on all the independent variables $U_j$, i.e., for $j = 1, 2, \ldots, m$, we fix $U_j = u_j$.

- When conditioning on $(X, Y) = (x, y)$, we simply use $\cdot|(X, Y)$.

- We assume $(U, V)$ is an independent copy of $(X, Y)$. Moreover, when conditioning on $(U, V) = (u, v)$, we simply use $\cdot|(U, V)$.

- $(a_1, a_2, \ldots, a_m)$ denotes the density argument for each dimension of $A$, and $(z_1, z_2, \ldots, z_K)$ denotes the density argument for each dimension of $Z$.

**Theorem 1.** *Given $A \sim RBG(X, \vec{\mathcal{V}}, \delta)$, the principal graph encoder embedding preserves the following conditional density:*

$$Y|A \overset{dist}{=} Y|Z \overset{dist}{=} Y|Z^D.$$

*Denote $L^*(Y, A)$ as the Bayes optimal error to classify $Y$ using $A$, we have*

$$L^*(Y, A) = L^*(Y, Z) = L^*(Y, Z^D).$$

*Proof.* The proof is decomposed into three parts:

- (i) establish $Y|A \overset{dist}{=} Y|Z$;

- (ii) establish $Y|Z \overset{dist}{=} Y|Z^D$;

- (iii) establish the Bayes error equivalence.

(i) It suffices to show the following always holds:

$$\text{Prob}(Y|A) = \text{Prob}(Y|Z)$$

where $Z = AW$ is the encoder embedding. Given that $Y$ is a categorical variable with prior probabilities $\{\pi_k, k = 1, \ldots, K\}$, each conditional probability satisfies

$$\text{Prob}(Y = y|A) = \frac{\pi_y f_{A|Y=y}(a_1, a_2, \ldots, a_m)}{\sum_{l=1}^{K} \pi_l f_{A|Y=l}(a_1, a_2, \ldots, a_m)},$$

$$\text{Prob}(Y = y|Z) = \frac{\pi_y f_{Z|Y=y}(z_1, z_2, \ldots, z_K)}{\sum_{l=1}^{K} \pi_l f_{Z|Y=l}(z_1, z_2, \ldots, z_K)}.$$

Therefore, it suffices to prove that the two numerators are proportional, i.e.,

$$c \times f_{A|Y}(a_1, a_2, \ldots, a_m) = f_{Z|Y}(z_1, z_2, \ldots, z_K)$$

for some positive constant $c$ that is unrelated to $Y$.

We begin by examining the conditional density of $A$:

$$f_{A|(Y,X,\vec{\mathcal{U}})}(a_1, a_2, \ldots, a_m) = \prod_{j=1}^{m} \delta(x, u_j)^{a_j} (1 - \delta(x, u_j))^{1-a_j}$$

$$= \prod_{k=1}^{K} \prod_{\substack{j=1,\ldots,m}}^{v_j=k} \delta(x, u_j)^{a_j} (1 - \delta(x, u_j))^{1-a_j}.$$

The first line follows because each dimension of $A$, under all the conditioning, is independently distributed as a Bernoulli random variable with probability $\delta(x, u_j)$ for $j = 1, \ldots, m$. Then the second line rearranges the product based on the class membership of each $v_j$.

We proceed by un-conditioning with respect to $\vec{\mathcal{U}}$, resulting in the following expression:

$$f_{A|(Y,X)}(a_1, a_2, \ldots, a_m) = \int_{\vec{\mathcal{U}}} f_{A|(Y,X,\vec{\mathcal{U}})}(a_1, \ldots, a_m) f_{\vec{\mathcal{U}}}(u_1, \ldots, u_m)$$

$$= \int_{u_1, \ldots, u_m} f_{A|(Y,X,\vec{\mathcal{U}})}(a_1, \ldots, a_m) f_{U_1}(u_1) \cdots f_{U_m}(u_m)$$

$$= \int_{u_1, \cdots, u_m} \prod_{k=1}^{K} \prod_{\substack{j=1,\ldots,m}}^{v_j=k} \delta(x, u_j)^{a_j} (1 - \delta(x, u_j))^{1-a_j} f_{U|V=v_1}(u_1) \cdots f_{U|V=v_m}(u_m)$$

$$= \prod_{k=1}^{K} \prod_{\substack{j=1,\ldots,m}}^{v_j=k} E(\delta(x, U)|V = k)^{a_j} (1 - E(\delta(x, U)|V = k))^{1-a_j}$$

$$= \prod_{k=1}^{K} (\tau_{x,k}(U))^{\sum_{j=1,\ldots,m}^{v_j=k} a_j} (1 - \tau_{x,k}(U))^{\sum_{j=1,\ldots,m}^{v_j=k} (1-a_j)}.$$

The first line is a standard application of conditional density manipulation, and note that $\vec{\mathcal{U}}$ is independent of $(X, Y)$. The second line rewrites the joint density of $f_{\vec{\mathcal{U}}}$ into individual densities, since the joint density is simply a product of $f_{U|V=v_j}(u_j)$. The fourth line computes the integral: since $u_j$ only appears once in the whole product, either via $\delta(x, u_j)^{a_j}$ or $(1 - \delta(x, u_j))^{1-a_j}$ due to $a_j$ taking values of either 0 or 1, solving the integral at each $j$ yields either $E(\delta(x, u_j))$ or $(1 - E(\delta(x, u_j)))$. Since this expectation is identical throughout the same $v_j$, we can represent this expectation as:

$$\tau_{x,k}(U) = E(\delta(x, U)|V = k).$$

This allows us to group terms with the same expectation together based on $k$.

Continuing with the derivation, we un-condition $X$ to derive $f_{A|Y}$:

$$f_{A|Y}(a_1, a_2, \ldots, a_m) = \int_x f_{A|(Y,X)}(a_1, a_2, \ldots, a_m) f_{X|Y}(x)$$

$$= \int_x \prod_{k=1}^{K} (\tau_{x,k}(U))^{\sum_{j=1,\ldots,m}^{v_j=k} a_j} (1 - \tau_{x,k}(U))^{\sum_{j=1,\ldots,m}^{v_j=k} (1-a_j)} f_{X|Y}(x).$$

Next, we consider the encoder embedding $Z$. Starting from $f_{Z|(Y,X,\vec{\mathcal{U}})}(z_1, z_2, \ldots, z_K)$, under such conditioning, the density at each dimension $k$ is a Poisson Binomial distribution, i.e.,

$$m_k Z_k | (Y, X, \vec{\mathcal{U}}) \sim \text{Poisson Binomial}(\{\delta(x, u_j)\})$$

for $j = 1, \ldots, m$ and $v_j = k$. After un-conditioning $\vec{\mathcal{U}}$, each probability $\delta(x, u_j)$ again becomes $E(\delta(x, U)|V = k)$ by the same reasoning as above for $f_{A|(Y,X)}$. Therefore,

$$m_k Z_k | (Y, X) \sim \text{Binomial}(m_k, \tau_{x,k}(U)).$$

As each dimension is conditionally independent, the density of $Z$ is the product of independent Binomials, and we have

$$f_{Z|(Y,X)}(z_1, z_2, \ldots, z_K) = \prod_{k=1}^{K} \binom{m_k}{m_k z_k} (\tau_{x,k}(U))^{m_k z_k} (1 - \tau_{x,k}(U))^{m_k - m_k z_k},$$

and

$$f_{Z|Y}(z_1, z_2, \ldots, z_K) = \int_x f_{Z|(Y,X)}(z_1, z_2, \ldots, z_K) f_{X|Y}(x)$$

$$= \int_x \prod_{k=1}^{K} \binom{m_k}{m_k z_k} (\tau_{x,k}(U))^{m_k z_k} (1 - \tau_{x,k}(U))^{m_k - m_k z_k} f_{X|Y}(x)$$

$$= \prod_{k=1}^{K} \binom{m_k}{m_k z_k} \int_x \prod_{k=1}^{K} (\tau_{x,k}(U))^{m_k z_k} (1 - \tau_{x,k}(U))^{m_k - m_k z_k} f_{X|Y}(x),$$

where the third line follows because $(m_k, z_k)$ are not affected by the integration of $x$.

Observe that the encoder embedding enforces $m_k z_k = \sum_{j=1,\ldots,m}^{v_j=k} a_j$ for each $k$. Comparing $f_{Z|Y}$ to $f_{A|Y}$, we immediately have

$$c \times f_{A|Y}(a_1, a_2, \ldots, a_m) = f_{Z|Y}(z_1, z_2, \ldots, z_K)$$

when $Z = AW$, where $c = \prod_{k=1}^{K} \binom{m_k}{m_k z_k}$ is a positive constant.

This conditional density equality holds regardless of the underlying $(X, \vec{\mathcal{V}})$ or $\delta(\cdot, \cdot)$. Hence, we have $Y|A \overset{dist}{=} Y|Z$ for the encoder embedding.

(ii) Without loss of generality, let us assume that $D = \{1, 2, \ldots, d\}$, and $d \in [1, K)$. This means the first $d$ communities are the principal communities, and the remaining are redundant communities. The trivial cases that $d = 0$ or $d = K$ will be addressed at the end

Recall the expression from part (i) above:

$$f_{Z|Y}(z_1, z_2, \ldots, z_K) = \prod_{k=1}^{K} \binom{m_k}{m_k z_k} \int_x \prod_{k=1}^{K} (\tau_{x,k}(U))^{m_k z_k} (1 - \tau_{x,k}(U))^{m_k - m_k z_k} f_{X|Y}(x).$$

This leads to:

$$\text{Prob}(Y = y|Z) = \frac{\pi_y f_{Z|Y=y}(z_1, z_2, \ldots, z_K)}{\sum_{l=1}^{K} \pi_l f_{Z|Y=l}(z_1, z_2, \ldots, z_K)}$$

$$= \frac{\pi_y \prod_{k=1}^{K} \binom{m_k}{m_k z_k} \int_x \prod_{k=1}^{K} (\tau_{x,k}(U))^{m_k z_k} (1 - \tau_{x,k}(U))^{m_k - m_k z_k} f_{X|Y=y}(x)}{\sum_{l=1}^{K} \pi_l \prod_{k=1}^{K} \binom{m_k}{m_k z_k} \int_x \prod_{k=1}^{K} (\tau_{x,k}(U))^{m_k z_k} (1 - \tau_{x,k}(U))^{m_k - m_k z_k} f_{X|Y=l}(x)}$$

$$= \frac{\pi_y \int_x \prod_{k=1}^{K} (\tau_{x,k}(U))^{m_k z_k} (1 - \tau_{x,k}(U))^{m_k - m_k z_k} f_{X|Y=y}(x)}{\sum_{l=1}^{K} \pi_l \int_x \prod_{k=1}^{K} (\tau_{x,k}(U))^{m_k z_k} (1 - \tau_{x,k}(U))^{m_k - m_k z_k} f_{X|Y=l}(x)}$$

where

$$\tau_{x,k}(U) = E(\delta(x, U^k)).$$

We first look at the terms from community $K$, which is assumed the redundant community. From the definition of redundant community, we have

$$\tau_{x,K}(U) = E(\delta(x, U^K)) = c_K$$

for all possible $x$ where $f_X(x) > 0$, where $c_K$ is a constant unrelated to $x$. Consequently, all terms involving $\tau_{x,K}(U)$ can be taken outside of the integral in both numerator and denominator, and the same holds for

terms associated with $\tau_{x,k}(U)$ for each $k = d+1, \ldots, K$. In essence, for any $l \in [1, K]$, we always have

$$\pi_l \int_x \prod_{k=1}^{K} (\tau_{x,k}(U))^{m_k z_k} (1 - \tau_{x,k}(U))^{m_k - m_k z_k} f_{X|Y=l}(x)$$

$$= (\prod_{k=d+1}^{K} c_k^{m_k z_k} (1 - c_k)^{m_k - m_k z_k}) \pi_l \int_x \prod_{k=1}^{d} (\tau_{x,k}(U))^{m_k z_k} (1 - \tau_{x,k}(U))^{m_k - m_k z_k} f_{X|Y=l}(x).$$

It follows that

$$\mathrm{Prob}(Y = y|Z) = \frac{\pi_y \int_x \prod_{k=1}^{d} (\tau_{x,k}(U))^{m_k z_k} (1 - \tau_{x,k}(U))^{m_k - m_k z_k} f_{X|Y=y}(x)}{\sum_{l=1}^{K} \pi_l \int_x \prod_{k=1}^{d} (\tau_{x,k}(U))^{m_k z_k} (1 - \tau_{x,k}(U))^{m_k - m_k z_k} f_{X|Y=l}(x)},$$

which exclusively pertains to dimensions corresponding to the principal communities. It is evident that:

$$\mathrm{Prob}(Y = y|Z^D) = \mathrm{Prob}(Y = y|Z) = \mathrm{Prob}(Y = y|A)$$

Hence, the principal graph encoder embedding satisfies $Y|A \stackrel{dist}{=} Y|Z^D$.

Regarding the two trivial cases: when $d = K$, implying that all communities are principal communities, the theorem trivially holds since no additional dimension reduction occurs. When $d = 0$, there is no principal community and $D$ is empty. In this scenario, we have

$$\mathrm{Prob}(Y = y|A) = \mathrm{Prob}(Y = y|Z) = \pi_y = \mathrm{Prob}(Y = y),$$

indicating that $A$ and $Y$ are independent, and $Z$ and $Y$ are independent as well. In other words, the graph provides no information for predicting $Y$, so the graph data itself is redundant.

(iii) Given two random variables $(X, Y)$ where $Y$ is categorical, the Bayes optimal classifier for using $X$ to predict $Y$ is

$$g(X) = \arg \max_{k=1,\ldots,K} \mathrm{Prob}(Y = k \mid X).$$

By the conditional density equivalence in parts (i) and (ii), it is immediate that the Bayes optimal classifier for using $A$ to predict $Y$ satisfies

$$g(A) = \arg \max_k \mathrm{Prob}(Y = k \mid A)$$
$$= \arg \max_k \mathrm{Prob}(Y = k \mid Z) = g(Z)$$
$$= \arg \max_k \mathrm{Prob}(Y = k \mid Z^D) = g(Z^D).$$

Therefore, the Bayes optimal classifier for predicting $Y$ is the same, regardless of whether the underlying random variable is $A$, $Z$, or $Z^D$. Since the optimal classifier is always the same, the resulting optimal error is also the same. $\square$

**Theorem 2.** *Assume $A \sim RBG(X, \vec{\mathcal{V}})$, and $\delta(X, U^k)|Y$ is independent of $X|Y$, which is satisfied under the stochastic block model. Then the population community score $\lambda(k) = 0$ if and only if community $k$ is a redundant community.*

*Proof.* (i) We first prove that the required condition, $\delta(X, U^k)|Y$ being independent of $X|Y$, can be satisfied under the stochastic block model (SBM).

Recall that the standard stochastic block model satisfies:

$$\mathbf{A}(i, j) \sim \mathrm{Bernoulli}(B(\mathbf{Y}(i), \mathbf{Y}(j))),$$

which, when cast into the framework of a random Bernoulli graph, is equivalent to:

$$\delta(X, U^k)|(Y = y) = B(y, k),$$

where $B(y, k)$ is a constant and, therefore, always independent of $X|(Y = y)$.

This condition can also hold under the more general degree-corrected stochastic block model (DC-SBM), with an additional assumption regarding how the degree parameters are generated. Under DC-SBM, we have:

$$\delta(X, U^k)|(Y = y) = \theta\theta' B(y, k),$$

where $\theta$ and $\theta'$ are the degrees for $X$ and $U^k$, respectively. Clearly, $\theta' B(y, k)$ is independent of $X|(Y = y)$. By further assuming that the degree variable $\theta$ is generated independently of $X|(Y = y)$, $\delta(X, U^k)|(Y = y)$ becomes independent of $X|(Y = y)$ under DC-SBM.

(ii) Next, we prove that under the condition that $\delta(X, U^k)|Y$ is independent of $X|Y$, the population community score $\lambda(k)$ equals 0 if and only if community $k$ is a redundant community.

From the definition of principal community and the population community score, we need to prove two things. First, we shall prove that

$$Var\left(E(\delta(X, U^k) \mid X)\right) = 0$$

if and only if

$$Var(E(Z_k|Y = l)) = 0.$$

This is because when the above conditional variance equals 0, $E(Z_k|Y = l)$ is a constant across different $l$, which makes the numerator of the population community score always 0.

From the proof of Theorem 1, we have:

$$m_k Z_k|(Y = l, X = x) \sim \text{Binomial}(m_k, E(\delta(x, U^k))).$$

From this, we can derive the conditional expectations as follows:

$$E(Z_k|(Y, X = x)) = E(\delta(x, U^k)),$$
$$E(Z_k|Y = l) = E(\delta(X, U^k)|Y = l).$$

When community $k$ is redundant such that

$$Var(E(\delta(X, U^k)|X)) = 0,$$

we immediately have

$$Var(E(Z_k|Y = l)) = Var(E(\delta(X, U^k)|Y = l)) = 0.$$

This is because when the conditional variance equals 0 for all possible $X$, it must also be 0 when conditioning on $Y = l$, which restricts to part of the support of $X$. This proves the only if direction.

To prove the if direction, we need the additional assumption that $\delta(X, U^k)|Y$ is independent of $X|Y$. Given such conditional independence, and the fact that $Z_k|Y$ is a random variable with parameter $m_k$ and $E(\delta(X, U^k))$, we immediately have that $Z_k|Y$ is independent of $X|Y$, which implies $E(Z_k|Y = l) = E(Z_k|(Y = l, X))$. Therefore, when $Var(E(Z_k|Y = l)) = 0$, we also have $Var(E(Z_k|(Y = l, X))) = 0$. Since $E(Z_k|(Y = l, X = x)) = E(\delta(x, U^k)) = E(\delta(X, U^k)|X = x)$, which intuitively means that conditioning on $Y$ is redundant once $X$ is known, this leads to $Var(E(Z_k|X)) = Var(E(\delta(X, U^k)|X)) = 0$.

(iii) Part (ii) proved that the numerator of the population community score equals 0 if and only if community $k$ is a redundant community. To complete the proof, it remains to show that the denominator of the population community score is greater than 0; otherwise, a 0/0 problem could arise when community $k$ is redundant.

Based on the binomial distribution of $Z_k$, we have:

$$Var(Z_k|(Y, X)) = \frac{E(\delta(x, U^k))(1 - E(\delta(x, U^k)))}{m_k} > 0,$$

which always holds, regardless of whether $k$ is redundant or not, except in the trivial case where $\delta(x, U^k) = 0$ or 1 almost surely. This corresponds to the scenario where the graph adjacency matrix is entirely 0s or 1s, making all communities redundant and reducing vertex classification to random guessing. Excluding such trivial cases, we have:

$$Var(Z_k|Y = l) = E_X(Var(Z_k|(Y, X))) + Var_X(E(Z_k|(Y, X))) > 0,$$

where the first term is positive, and the second term is non-negative.

Thus, excluding trivial graphs, the denominator of the principal community score is always positive, regardless of whether community $k$ is redundant or principal. $\qquad\square$

