# OpenReview forum: "Principal Graph Encoder Embedding and Principal Community Detection"
_TMLR — Rejected by TMLR_

### Review · Reviewer_29ug · 2024-07-03

**Summary Of Contributions:**

The paper introduces the concept of principal communities within graph data and proposes a novel principal graph encoder embedding method. This method seeks to concurrently detect these principal communities and achieve vertex embedding. Using a graph adjacency matrix with vertex labels, the method computes a sample score for each community to rank their importance and estimate the principal communities. It then retains only the dimensions corresponding to these principal communities for the vertex embedding. Theoretical properties under the random graph model are characterized, and simulations and real-data experiments are conducted to show the method's performance.

**Audience:**

Yes

**Broader Impact Concerns:**

There are no concerns regarding the ethical implications of this work that would require adding a Broader Impact Statement.

**Claims And Evidence:**

Yes

**Requested Changes:**

1. Include a dedicated section that explicitly details the differences between the one-hot graph encoder embedding and the proposed principal graph encoder embedding. Highlight what new problems are being solved, how the proposed method improves upon the existing one, and the specific motivation for developing this variant.

2. Clearly illustrate the benefits of the proposed method over the original one-hot graph encoder embedding. If the performance metrics are similar, focus on other aspects such as computational efficiency, scalability, or robustness. Please provide sufficient empirical evidence for these advantages.

3. Expand the literature review to better position the proposed method within the existing body of work. Discuss how it compares to other graph embedding and community detection techniques, not just the one-hot graph encoder embedding, and highlight its unique contributions.

4. Clearly articulate the differences in theoretical results between the proposed method and the original method. Additionally, provide a discussion on how the theoretical properties at the population level translate to the sample level, and present empirical or theoretical evidence to support them.

**Strengths And Weaknesses:**

## Strengths

1. The method is built on a robust theoretical framework, ensuring that it retains sufficient information about vertex labels and can be Bayes optimal for vertex classification.

2. The paper includes extensive simulations and real-data experiments that provide strong empirical support for the method's effectiveness.

3. The method’s ability to preserve the conditional density of the label vector is a significant theoretical strength, ensuring optimal classification performance.

## Weaknesses

1. The paper builds on the one-hot graph encoder embedding method but does not clearly articulate the differences between the two methods. It’s crucial to specify what issues the proposed method addresses that the original does not, and what advantages the proposed method has. The motivation for developing this variant method is not sufficiently explained. The paper should provide a strong rationale for why this new approach is necessary and what specific issues it addresses.

2. The performance of the proposed method is reported to be close to the original one-hot graph encoder embedding method, as shown in Table 1. The paper needs to clearly demonstrate the advantages of the proposed method despite the similar performance metrics, whether in terms of computational efficiency, scalability, or other qualitative aspects.

3. The literature review does not adequately position the proposed method in relation to existing graph embedding and community detection techniques. A more detailed comparison with related work is necessary to highlight its unique contributions.

4. The difference in theoretical results between the proposed method and the original method is not clearly articulated. Additionally, the current theoretical results are presented at the population level. Since the theoretical framework is based on a probabilistic model, the authors should also consider results at the sample level to ensure practical applicability.

---

> ### Author Response · Authors · 2024-08-23
>
> Thank you for reviewing our paper and for the valuable suggestions! We have made significant updates in the revised paper, including a stronger emphasis on the contributions of the new method, its computational advantages, the differences between P-GEE and the original GEE, a new theoretical section on the sample and population community scores, and a new noisy real-data section to better illustrate the numerical advantages of the proposed method.
>
> Indeed, the main contribution lies in steps 5 and 6, where the sample community score is computed and the dimensionality is subsequently reduced. To address the review concern and to strengthen the contributions, we have made the following changes:
>
> 1. Enhanced Method Description: The last three paragraphs of Section 1, on page 2, provide a more comprehensive description of the new method and the advantages of additional dimension reduction. Additionally, at the end of Section 2.1 on page 3, we emphasized that the main contribution of the method is in steps 5 and 6, which involve the computation of the sample community score and subsequent dimension reduction.
>
> 2. Theoretical Justification: To better connect the sample method with the population theory, particularly to address the lack of justification for the sample community score, we added a new Section 3.5 and Theorem 2. Specifically, we now show that the sample community score converges to the population community score (this follows immediately since the sample statistic uses the sample quantities of the population version). Furthermore, Theorem 2 demonstrates that the population community score equals 0 if and only if community k is redundant. Theorem 2 holds under a technical condition that is satisfied by the SBM and DC-SBM models.
>
> With the introduction of Theorem 2, we have closed an important gap and shown that the proposed method indeed detects the defined principal communities. Additionally, Theorem 1 demonstrates that the embedding preserves the conditional density after excluding these defined principal communities.
>
> Following Theorem 2, we also provide an important discussion on alternative statistics that can consistently detect principal communities, as well as the impact of sample estimation variance, which suggests that the results hold for sample graphs when the vertex size is sufficiently large.
>
> 3. Numerical Illustration: To better illustrate the numerical advantages, we introduced a new noisy-data setting on page 11, where 10\% of the vertex labels are randomly assigned to 30 additional noise classes. Table 2 on page 13 reports the results, showing that the proposed P-GEE is more robust and significantly better in classification error under these noisy conditions. Moreover, the dimension reduction via the community score successfully recovers the ground truth in most cases.
>
> 4. Computational Advantage: We discussed the computational advantages in Section 2.5. Table 2 on page 13 further confirms this analysis, showing that vertex classification using P-GEE is faster than GEE due to the reduced dimensionality (in this case, the removal of 30 noise dimensions). Moreover, both P-GEE and GEE are much faster than spectral embedding.
>
> It is important to note that the reported running time for spectral embedding is actually an understatement, as real graphs are almost always stored in edgelist format. GEE is directly applicable to edgelists, while spectral embedding and many existing graph embedding approaches require a transformation from edgelist to adjacency matrix, incurring an additional computational burden. For example, in the case of the Protein data, ASE required 2.4 seconds, but there was a hidden cost of 10 seconds to transform the edgelist into a sparse adjacency matrix.

---

### Review · Reviewer_P9NC · 2024-08-02

**Summary Of Contributions:**

The paper proposes a graph node embedding particularly focusing on preservation of community information. The authors employ a simple embedding procedure called principal graph encoder, defined by a product of adjacency matrix and a (weighted) class label matrix. For the embedding, the authors introduce 'community score' of each class. If the community score is larger than the given threshold, that community is regarded as a 'principal community'. Further, theoretical analysis about the conditional distribution preservation is considered in the 'population' setting.

**Audience:**

Yes

**Broader Impact Concerns:**

none.

**Claims And Evidence:**

Yes

**Requested Changes:**

One of major claims is to achieve 'further dimension reduction' over original graph encoder embedding (described in introduction). However, a rationale behind the proposed community score based dimension reduction (step5) is quite unclear to me. Interpretation of the definition of the community score \hat{\lambda}(k) is not clarified in detail. What does this equation represent? Why this quantity is effective to evaluate the score of k-th community? Please clarify in more detail.

Please clarify implication of theorem 1, particularly about concerns I mentioned above in W4.

**Strengths And Weaknesses:**

S1: The graph embedding is obviously important topic.

S2: Considering evaluation of principal community is seemingly interesting.

W1: As the authors mentioned, Steps 1 to 3 of the algorithm are the same as in existing papers. The differences lie in Step 4's normalization and Step 5's community score. The normalization in Step 4 is quite simple, and it is hard to say that it has any novelty (almost same normalization is performed in existing papers).

W2: Validity of the community score is unclear. Additionally, the theoretical analysis does not depend on the community score (and normalization), so the theoretical analysis is not a justification of the new methodology. I find this misleading.

W3: It is unclear what the theoretical analysis in this paper implies. The analysis is done in the context of the population, but it deviates from the actual algorithm, raising doubts about whether the actual procedure is justified in any way.

W4: Theorem 1 deviates from actual situations, and it is unclear what significance there is in considering population estimation defined in the theorem.
- In Theorem 1, the true D, the index set of principal communities, is assumed to be known, though it is unknown. Therefore, Theorem 1 does not justify the proposed community score and its thresholding.
- Theorem 1 assumes that Z is implicitly determined from the true Y (which does not take '0') according to the definition Z = A W. However, in the case of community detection, W is likely to be mostly missing in samples (i.e., Y would take many 0) for the purpose of community detection.
- Theorem 1 does not normalize Z, while the algorithm normalizes it.

W5: There are already many papers (eg [1-3] and other papers by the same author) based on almost the same methodology (except for community score). In my understanding, the main novel contents of this paper are community score and the theorem. The qualitative validaty of the community score and implication of theorem are unclear as already mentioned. I question whether this paper contains substantially informative and different contents of being published as a new paper.

[1] Shen et al., IEEE TNSE 2024
[2] Shen et al., International Conference on Algorithms, Data Mining, and Information Technology, 2023
[3] Shen et al., IEEE TPAMI 2023

---

> ### Author Response · Authors · 2024-08-23
>
> Thank you for raising this important concern regarding the method and the sample community score.
>
> Indeed, the main contribution lies in steps 5 and 6, where the sample community score is computed and the dimensionality is subsequently reduced. To address this concern and strengthen the theoretical foundation, we have made the following changes:
>
> 1. Emphasis on Main Contribution: At the end of Section 2.1 on page 3, we have emphasized that the main contribution of the method is in steps 5 and 6.
>
> 2. Theoretical Justification: To better connect the sample method with the population theory, particularly to address the lack of justification for the sample community score, we have added a new Section 3.5 and Theorem 2. Specifically, we now show that the sample community score converges to the population community score (this follows directly since the sample statistic uses the sample quantities of the population expectation / variance). Moreover, Theorem 2 proves that the population community score equals 0 if and only if community k is redundant. Theorem 2 holds under a technical condition that can be satisfied by the SBM and DC-SBM models.
>
> We believe that the introduction of Theorem 2 closes this important gap and demonstrates that the proposed method effectively detects the defined principal communities. Additionally, Theorem 1 shows that the embedding preserves the conditional density after excluding these defined principal communities.
>
> Following Theorem 2, we also provide an important discussion on alternative statistics that can consistently detect principal communities. We chose the current sample score because it is both theoretically sound and empirically well-behaved, as evidenced by Figures 1 and 2, which successfully separate principal and redundant communities in simulated sample graphs.
>
> 3. Demonstration of Numerical Performance: To further illustrate the numerical performance, we introduced a new noisy-data setting in the revised draft on page 11. In this setting, 10\% of the vertex labels are randomly assigned to 30 additional noise classes. Table 2 on page 13 reports the results, showing that the proposed P-GEE significantly improves classification error. Moreover, the dimension reduction via the community score successfully recovers the ground truth in most cases.

---

### Review · Reviewer_pse1 · 2024-08-17

**Summary Of Contributions:**

The paper introduces a novel method P-GEE for detecting principal communities and achieving vertex embedding in graphs. The proposed method has sound theoretical properties to guarantee preservation sufficient information about the vertex labels.

**Audience:**

Yes

**Claims And Evidence:**

Yes

**Requested Changes:**

See above

**Strengths And Weaknesses:**

Strengths:

Novel Methodology: The paper introduces a new approach to community detection and graph embedding by focusing on principal communities, offering a fresh perspective in the field.
Theoretical Contribution: The authors provide a theoretical framework for the proposed method, demonstrating its ability to preserve sufficient information about vertex labels, adding rigor to the proposed method.
Flexible Application: The proposed method can be applied to both labeled and non-labeled graphs, making it useful for both supervised and unsupervised graph-based applications.
Weaknesses:

Limited Scope of Application: The paper focuses primarily on graph embedding methods, which may have limited application compared to GNNs that utilize node features for richer representations.
Experimental Rigor: The experiments conducted in the paper are not comprehensive enough to fully validate the method's claims. The authors rely on visualization and results from relatively small datasets, which may not sufficiently demonstrate the method's robustness and scalability.
Readability Issues: The paper's readability could be improved, particularly in terms of equation labeling and overall presentation. Also, there is a typo in the explanation of the Degree-Corrected Stochastic Block Model (Sec 3.1). The correct notation should be
 instead of
.

---

> ### Author Response · Authors · 2024-08-23
>
> Thank you for reviewing the paper and providing constructive feedback.
>
> To further strengthen the paper and better distinguish the advantages of the proposed method, we have made the following updates:
>
> 1. Computational Complexity Analysis: On page 4, we added an analysis showing that the computational complexity is linear with respect to the number of vertices. The running time advantage is further confirmed in Table 2 on page 13.
>
> 2. New Theoretical Subsection: On page 8, we introduced a new subsection where Theorem 2 proves the theoretical properties of the community score, bridging a theoretical gap in the initial submission.
>
> 3. Noisy-Data Setting: On page 11, we added a new setting where 10% of the vertex labels are randomly assigned to 30 additional noise classes. Table 2 on page 13 reports the results, demonstrating that the proposed P-GEE significantly improves classification error. Additionally, the dimension reduction via the community score successfully recovers the ground truth in most cases, and the running time using P-GEE can be better than GEE due to the reduced dimensions, and much better than spectral embedding.
>
> We have also made minor tweaks throughout the paper to improve presentation.
>
> Please note that while this paper does not address node features, which are not always present in graph data, it is methodologically possible to extend the embedding to include them, similar to GCN. However, the theory in this case would be more complex and notation-heavy, as it would require consideration of latent positions, graph variables, and standard node feature variables. This extension is currently under development and falls outside the scope of the present paper.

---

### Decision · Action_Editor_RChW · 2024-11-25

**Recommendation:** Reject

**Comment:**

A significant portion of the proposed approach has appeared in prior work. Furthermore, the authors have only partially answered the questions raised in the reviews (such as by introducing Theorem 2). For the theoretical part, I think there is also some lack of formality. For instance Definition 1 (which is very long), starts as "Given a vertex, we assume Y .. ". While I understand what they mean, this is not how one would introduce a new probabilistic model, especially if the intention is to prove results such as Theorem 1. As it stands I'm not convinced that all claims are completely accurate. I concur with the reviewer recommendations and recommend rejection. The authors are encouraged to revise carefully, making definitions more formal, addressing all the comments in the reviews. A major revision could be reconsidered by TMLR.

**Audience:**

Community detection and graph embeddings are important subfields of Machine Learning. There would be a wide audience for a paper on these topics.

**Claims And Evidence:**

The paper is about graph embeddings, particularly for graphs with some community structure. The paper introduces the notion of principal communities and proposes methods to detect these and perform a suitable vector embedding. The paper has some theoretical results as well as some empirical studies. There are some gaps between what the theory suggests and the experiments.

**Resubmission Of Major Revision:**

The authors may consider submitting a major revision at a later time.